# Emotional Intelligence, Physical Activity Practice and Mediterranean Diet Adherence-An Explanatory Model in Elementary Education School Students

**DOI:** 10.3390/children9111770

**Published:** 2022-11-18

**Authors:** Eduardo Melguizo-Ibáñez, Gabriel González-Valero, Pilar Puertas-Molero, José Manuel Alonso-Vargas

**Affiliations:** Faculty of Education Sciences, Department of Didactics of Musical, Plastic and Corporal Expression, University of Granada, 18071 Granada, Spain

**Keywords:** healthy lifestyle, teenagers, active lifestyle, elementary students

## Abstract

Currently, there is a global concern with regard to the lifestyles of young people. This study aims to study the association between Mediterranean diet adherence, emotional intelligence and physical activity practice in teenagers in the last cycle of elementary education. In turn, this objective is divided into (a) developing an explanatory model of the practice of physical activity, Mediterranean diet adherence and emotional intelligence; and (b) developing a multi-group model according to the gender of the participants. A descriptive, cross-sectional, comparative study was conducted in a sample of 293 elementary school students (M = 11.45; S.D = 0.31). The instruments used were an ad hoc socio-demographic questionnaire, the Trait Meta Mood Scale-24, the KIDMED questionnaire and the Physical Activity Questionnaire for Older Children. The results show that males show a positive relationship between adherence to the Mediterranean diet and emotional intelligence and between emotional intelligence and physical activity. In contrast, in the case of females, a negative relationship was observed between emotional intelligence and physical activity. In conclusion, it can be seen that gender plays a fundamental role in adolescence, being a key factor influencing an active and healthy lifestyle.

## 1. Introduction

Adolescence occurs between childhood and adulthood [1], with crucial physical, psychological, and social changes [2]. Both development and growth are supposed to be two processes that occur with a high degree of intensity with adolescence, as both are the result of an interaction between environmental and genetic factors [3]. In this case, development during adolescence is influenced by the gender of the participants [4]. Numerous national and international studies [5,6] have found statistically significant differences in gender differences when it comes to the pursuit of an active and healthy lifestyle. In this case, it has been found that males are more likely to engage in more weekly physical activity during adolescence, with females showing higher levels of sedentary behaviour [7]. In contrast, it has been observed that males show higher levels of emotional competence and greater adherence to a healthy dietary pattern [8].

Nutrition is supposed to play a fundamental role in the development and growth of different subjects [3]; however, during this stage, due to the biological, psychological and social changes that take place, human growth and development can be affected in a negative way [9]. In view of this problem, the Mediterranean diet is considered a healthy dietary model due to its contribution to optimal health and quality [10]. This dietary pattern is characterised by a wide variety of foods, such as wholegrain cereals, olive oil, bread, milk and dairy products, fruit, vegetables, and nuts [11]. Optimal adherence to the Mediterranean diet offers numerous benefits, such as reduced waist circumference, appropriate body fat percentage, increased life expectancy [12], as well as a reduced likelihood of cardiovascular disease, neurodegenerative diseases, and various types of cancer [13,14]. It has been observed that during adolescence there is a detachment from this dietary pattern [15], especially in males, as young people begin to have more control over their dietary pattern [14]. Following this type of diet not only has a positive effect on health, but also has a beneficial influence on other areas such as the control of emotions [15,16].

Emotional competence plays a fundamental role in people’s lives, as it equips individuals with the necessary skills to cope with different situations, as well as to promote people’s psychological well-being, regardless of age [17,18]. Likewise, being emotionally competent helps to improve concentration and the control of stressful situations, as well as self-motivation [19], having a positive impact on people’s mental health. The concept of emotional intelligence is derived from emotional competence, which is conceived as a construct that constitutes the psychological development of emotions [17,20], and is composed of attention to these states, clarity of understanding, and the repair of negative emotions [21], which help to control emotions [21].

Emotional control plays a fundamental role in leading a psychologically healthy lifestyle [2], and is related to emotional intelligence [21]. In this case, emotional control is seen as a multidimensional experience with three response systems: cognitive/subjective, behavioural/expressive, and physiological/adaptive [22]. In the educational setting, emotional intelligence is conceived as a key element for emotional control [23]. Within the educational context, the model proposed by Salovey et al. [22] is one of the most studied, as it conceives of emotional intelligence as a three-dimensional construct composed of three areas: emotional perception, emotional clarity and emotional repair. During adolescence, numerous emotional ups and downs have been observed, caused by different behaviours, predominantly those originating in the social area [24]. The study by Sanchís-Sanchís et al. [24] affirms that females attach greater importance to emotional control than males during this stage of development, and that this control leads to an improvement in educational, social and mental well-being [17,18].

Another key factor in achieving mental well-being is physical activity [25]. In this case, during adolescence, there is a decline in weekly physical activity time, as young people opt for other types of more sedentary tasks [26]. In view of this information, an increase in cardiovascular diseases has been observed in the youth population [27], which is detrimental to the public health of young people. Furthermore, it has also been observed that low levels of physical activity have been reported to lead to a poorer perception of people of themselves and of their physical and social abilities [28]. During adolescence, research conducted by Vaquero-Solís et al. [28] states that girls show a longer sedentary lifestyle than boys, and these results offer a problem that needs to be addressed from the educational sphere, specifically from the area of physical education.

That is why the following study shows the following research hypotheses:

**H.1.** *Adolescent students will show a positive link between physical activity and emotional intelligence and Mediterranean diet adherence*.

**H.2.** *Male participants will show a better relationship between adherence to the Mediterranean diet and physical activity than females*.

**H.3.** *Male participants will show a worse relationship between adherence to the Mediterranean diet and emotional intelligence than females*.

**H.4.** *Male participants will demonstrate a worse relationship between adherence to the Mediterranean diet and physical activity than females*.

**H.5.** *Female participants will obtain a better relationship between adherence to the Mediterranean diet and emotional intelligence than males*.

The aim of the research is to establish the relationship between adherence to the Mediterranean diet, emotional intelligence and physical activity practice in adolescents in the third cycle of primary education, broken down into (a) developing an explanatory model of physical activity practice, adherence to the Mediterranean diet and emotional intelligence; and (b) developing a multi-group model regarding the gender of the sample.

## 2. Materials and Methods

### 2.1. Design and Participants

A comparative, descriptive, and cross-sectional research design was used. The sample was comprised of a total of 293 elementary school students, aged between 11 and 12 years (M = 11.45 ± 0.31). For data collection, convenience sampling was carried out, inviting to participate those students who met the inclusion criteria. In this case, the criteria for participation consisted of being under 12 years of age and being in the third cycle of elementary education. In this case, the exclusion criteria established were as follows: not belonging to the third cycle of the primary education stage and not being aged between 11 and 12. Therefore, the inclusion criteria were to be aged between 11 and 12 years and to be in the last stage of primary education. Failure to meet the aforementioned criteria meant direct exclusion from the study. The distribution of the sample is homogeneous, with 50.2% of the sample being male (n = 147) and 49.8% female (n = 146). In terms of sampling error, for a maximum error for a confidence level of 95%, an error of 4.19% was achieved.

### 2.2. Instruments

**Sociodemographic questionnaire** was designed to collect the socio-demographic variables (sex and age). In this case, for the collection of the sex variable, the options of “Male” and “Female” were presented, while for age, no option was offered, but rather a free response was given by the participants.

**Trait-Meta-Mood Scale (TMMS-24)** was used for collecting the emotional intelligence variable. The instrument was developed by Salovey et al. [29], however, due to the characteristics of the sample, the version of Fernández-Berrocal et al. has been used [30]. The instrument is made up of 24 items. This questionnaire conceives emotional intelligence as a three-dimensional variable where emotional intelligence is assessed through three sub-variables: emotional attention, emotional clarity and emotional repair. To study the reliability of the results, the Cronbach’s Alpha test was used, obtaining the following scores: emotional attention (EA) (α = 0.845), emotional clarity (EC) (α = 0.800), and emotional repair (ER) (α = 0.863).

**KIDMED questionnaire** was used to assess the degree of Mediterranean Diet adherence. This study has used the version developed by Serrá-Majem et al. [31]. The questionnaire is composed of 16 questions which are answered positively or negatively. Depending on the participants’ answers, the final score of the questionnaire varies between −4 points and 12 points. The instrument categorizes the degree of adherence to the Mediterranean Diet into three levels: optimal diet (≥8 points), needs improvement (2–7 points), and poor-quality diet (≤1). A reliability analysis yielded a value of α = 0.716.

**Physical Activity Questionnaire for Older Children (PAQ-C),** which is validated for the population aged between 8 and 14 years [32], was used to measure the physical activity variable. The Spanish version of Manchola-González et al. [33] was used. This instrument is made up of a total of 10 items that are scored from 1 to 5. The questionnaire showed a reliability of α = 0.743.

### 2.3. Procedure

Before starting data collection, a literature review and a systematic review were carried out to gain a better comprehension of the problems addressed. Subsequently, we proceeded to search for instruments validated by the scientific community, obtaining those described above. Once the questionnaire had been prepared, we proceeded to contact the different educational institutions, guaranteeing the anonymity of each of the participants. Once the different schools had authorized the researchers’ entry, the legal guardians of the minors were contacted. They were informed of the different objectives of the study and were assured that the data would be processed for scientific purposes. Likewise, having obtained the consent of the legal guardians, the data were collected, and the researchers were present during the collection of the data to resolve any doubts that may have arisen. Finally, the present study followed the principles established in the Helsinki Declaration of 1975. Likewise, this study was supervised by the Ethics Committee of the University of Granada (2966/CEIH/2022).

### 2.4. Data Analysis

IBM SPSS Amos 26.0 (IBM Corp, Armonk, NY, USA) was used to propose the structural equation models. This type of analysis makes it possible to study the effect of the proposed variables among the general study population, both male and female. Taking into account the characteristics of each of the models proposed, each one is made up of a total of four endogenous variables (EA; EC; ER; PA) and two exogenous variables (MDA; EI) (Figure 1). For the endogenous variables, and the causal associations were examined on the basis of the observed associations between the indicators and the degree of reliability of the measurements. This allows for the inclusion in the model of the measurement error of the observed variables. The one-way arrows represent the lines of influence and are produced from the regression weights. Likewise, two levels of significance have been established, one for *p* ≤ 0.05 and the other for *p* ≤ 0.001. It was observed that the models proposed established relationships between emotional intelligence and the physical-health variables, showing relationships between the two.

In order to assess the fit model, the recommendations proposed by Bentler [34] and McDonald and Marsh [35] have been followed. The model should be fitted by the following fit indices: Goodness of Fit Index (GFI), Comparative Fit Index (CFI) and Incremental Reliability Index (IFI), all of which reflect scores above 0.900 for a good fit. Another dimension to take into account is the Root Mean Square Approximation (RMSEA), reflecting values below 0.100 for a good fit.

**Figure 1 children-09-01770-f001:**
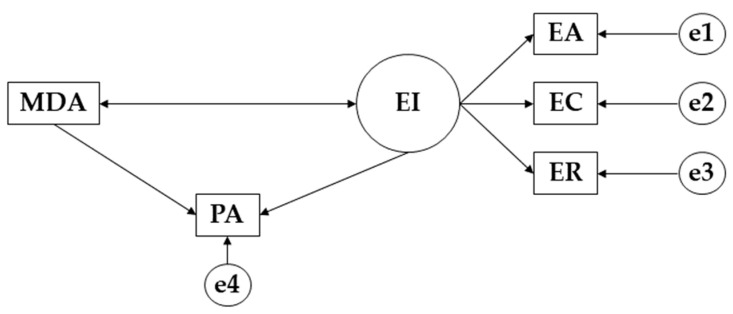
Structural equation model. **Note:** Emotional Clarity (EC); Emotional attention (EA); Emotional Repair (ER); Mediterranean diet adherence (MDA); Physical activity (PA).

## 3. Results

The structural equation model proposed for the whole sample shows a good fit for each of its indices (Figure 2). The Chi-Square analysis obtained a significant value (X^2^ = 1.198; df = 4; pl = 0.878). Despite these good results, an isolated interpretation is not possible due to the sample size and statistical sensitivity of the sample, which is why other standardized adjustment indices have been used [36]. In this case, the comparative fit index (CFI), normalized fit index (NFI), incremental fit index (IFI) and Tucker Lewis index were higher than 0.900, while the root mean square error of approximation (RMSEA) showed a value of 0.009.

Figure 2 and Table 1 show the associations for the total study population. Positive relationships are observed between emotional attention (EA) (r = 0.537), emotional clarity (EC) (*p* ≤ 0.001; r = 0.735) and emotional repair (ER) (*p* ≤ 0.05; r = 0.771) with respect to emotional intelligence (EI). Regarding the relationship between Mediterranean Diet adherence (MDA) and physical activity (PA), positive relationships were observed (*p* ≤ 0.05; r = 0.143), with a positive relationship between the practice of physical activity (PA) and emotional intelligence (EI) (r = 0.091). A positive link was obtained between Mediterranean Diet Adherence (MDA) and emotional intelligence (EI) (*p* ≤ 0.05; r = 0.223).

The proposed model’s male population shows a good fit for each of its indicators. The Chi-Square analysis shows a significant value (X^2^ = 4.130; df = 4; pl = 0.389). The comparative fit index (CFI), normalized fit index (NFI), incremental fit index (IFI) and Tucker Lewis index (TLI) showed values of 0.999, 0.959, 0.999 and 0.996, respectively. On the other hand, the RMSEA showed a score of 0.015.

Table 2 and Figure 3 show the relationships for the males. In this case, positive relationships are obtained between emotional attention (EA) (r = 0.421), emotional clarity (EC) (*p* ≤ 0.001; r = 0.807) and emotional repair (ER) (*p* ≤ 0.001; r = 0.668). Likewise, a positive connection is also shown between emotional intelligence (EI) and physical activity (PA) (*p* ≤ 0.05; r = 0.363), with exactly the same occurring between Mediterranean Diet adherence (MDA) and emotional intelligence (EI) (*p* ≤ 0.05; r = 0.336). Finally, a negative relationship was found between Mediterranean Diet adherence (MDA) and physical activity (PA) (r = −0.069).

The model developed for the female population obtained a good fit for its indices. In this case the Chi-Square analysis obtained a significant value (X^2^ = 8.564; df = 4; pl = 0.073). The comparative fit index (CFI), normalized fit index (NFI), incremental fit index (IFI) and Tucker Lewis index (TLI) obtained values of 0.964, 0.937, 0.966 and 0.910 respectively, while the RMSEA score was 0.088.

Figure 4 and Table 3 show the effects of the variables developed for females. In this case, it was observed that there is a positive relationship between emotional intelligence and the three component variables (EA, r = 0.645; EC *p* ≤ 0.001, r = 0.750; ER, *p* ≤ 0.001, r = 0.783). Likewise, a positive relation is observed between physical activity (PA) and Mediterranean Diet adherence (*p* ≤ 0.001; r = 0.272). The same happens for the variables Mediterranean Diet adherence (MDA) and emotional intelligence (EI) (r = 0.127). A negative association was found between physical activity (PA) and emotional intelligence (EI) (r = −0.044).

## 4. Discussion

The current research proposes a structural equation model between Mediterranean Diet adherence, emotional intelligence and the practice of physical activity in adolescents in the last stage of elementary education. The aim of this discussion is to compare the results obtained with those of other studies already carried out.

In this case, the model proposed for the sample under study shows a positive relationship between Mediterranean Diet adherence and physical activity practice. Trigueros et al. [37] state that the subject of physical education helps the physical and healthy development of the adolescent population. Likewise, Melguizo-Ibáñez et al. [38] together with Villodres et al. [10] establish that the socioeconomic level of families also influences young people’s active and healthy lifestyles, with parents being a fundamental element in promoting these lifestyles. Positive relationships have also been found between physical activity and emotional intelligence. In view of these findings, the study by Melguizo-Ibáñez et al. [38] and López-Sánchez et al. [39] affirm that regular physical exercise promotes the secretion of neurotransmitters, which help to channel disruptive states. Furthermore, Duclos-Bastías et al. [40] state that physical exercise helps to improve young women’s self-image, reporting benefits in self-concept. A positive link is observed between emotional intelligence and Mediterranean Diet adherence. The study by Pérez-Mármol et al. [41] affirms that a healthy diet together with an active lifestyle brings benefits in self-concept, with improvements in the physical and emotional areas of young people.

In the model proposed for male adolescents, a negative relationship is observed between physical activity and Mediterranean Diet adherence. Very different results have been found by Franco et al. [42]; however, Moschonis et al. [43] establish that inadequate nutritional education on the part of young people leads to a worsening of the dietary pattern. To alleviate this poor training, Ruiz et al. [44] suggest the presence of a greater number of hours of physical education through which young people can be trained from a dual perspective, highlighting the active and healthy aspect of this subject. On the other hand, a positive relationship is obtained between the practice of physical activity and emotional intelligence. Very different results were discovered by Zurita-Ortega et al. [45], who found that that young people tend to orient sport practice towards the ego climate, and when the proposed objectives are not achieved, this leads to an increase in the levels of frustration and anxiety. Similarly, a positive relationship is observed between adherence to the Mediterranean Diet and emotional intelligence. In view of these results, Melguizo-Ibáñez et al. [38] establish that a healthy lifestyle helps to improve physical appearance and thus gives rise to emotional satisfaction derived from young people’s perception of their bodies.

Regarding the model developed for the female population, a negative association between physical activity and emotional intelligence is evident. These findings coincide with the assertion made by Portela-Pino et al. [46], where they state that the decrease in weekly physical activity times encourages the appearance of disruptive states that affect the physical and psychological well-being of young people. Likewise, Zhang et al. [47] affirm that during adolescence there is a reduction of weekly physical activity times, as adolescents give greater priority to other types of tasks with a more sedentary character. In contrast, a positive relationship is observed between adherence to the Mediterranean diet and physical activity. In view of this statement, Ahmad-Bahathig et al. [48] state that there is an awareness among adolescents of the health benefits of physical activity and the Mediterranean Diet in terms of an active and healthy lifestyle. Likewise, it is also observed that there is a positive relationship between Mediterranean Diet adherence and emotional intelligence, coinciding with the results of López-Olivares et al. [49], who claim that a healthy diet helps to improve people’s physical and mental health.

The study by Melguizo-Ibáñez et al. [16] found that the degree of family functioning helps in the development of positive adherence to a healthy dietary pattern as well as in the acquisition of greater emotional competence. Regarding these findings, Wang et al. [50] affirm the importance of physical education in the school curriculum, as they state that through physical sports practice a regulation process is carried out through which disruptive states are channeled. Furthermore, it has also been shown that not just any type of physical activity brings benefits in the emotional area, as research by Skurvydas et al. [51] shows that improvements at the emotional level are observed when physical activity is carried out at a moderate or vigorous level.

## 5. Limitations and Strengths

The present study, despite having responded to the objectives initially set out, has a series of limitations which are highlighted below. The first of these lies in the nature of the study, due to the design only allowing for the study of cause-effect relationships at that point in time. With respect to the sample, it does not allow for generalizations to be established in a bigger area of the national geography. Similarly, the instruments used, despite having been validated by the scientific community, show an intrinsic error in the measurement of the data.

In terms of the strengths of this research, the type of analysis used stands out. In this case, this type of analysis makes it possible to study the effect of the variables on each other according to the direction of the proposed arrows.

## 6. Conclusions

The model developed for the sample of adolescents shows positive relationships between physical activity and Mediterranean Diet adherence. In addition, a positive relation is observed between emotional intelligence and physical activity, with exactly the same occurring between emotional intelligence and Mediterranean Diet adherence.

The model proposed for males shows a negative relationship between Mediterranean diet adherence and physical activity practice. Positive relationships are observed between physical activity practice and emotional intelligence, as well as between Mediterranean Diet adherence and emotional intelligence.

The model developed for the female population shows evidence of a negative relation between emotional intelligence and physical activity. In contrast, a positive relationship is observed between physical activity and Mediterranean Diet adherence. Likewise, a positive link between emotional intelligence and Mediterranean Diet adherence was determined

Finally, the present study highlights that sex plays a fundamental role in adolescence, it being a key factor that influences an active and healthy lifestyle. It is for this reason that special emphasis should be placed on transmitting healthy lifestyles and educating young people about emotions in education.

## Figures and Tables

**Figure 2 children-09-01770-f002:**
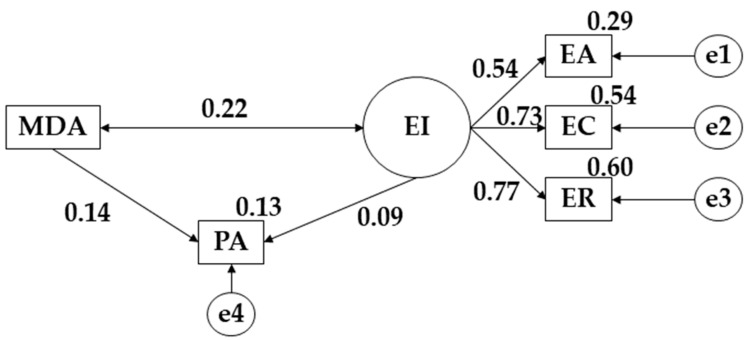
Model proposed for the entire sample. **Note:** emotional clarity (EC); emotional repair (ER); emotional attention (EA); Mediterranean Diet adherence (MDA); physical activity (PA).

**Figure 3 children-09-01770-f003:**
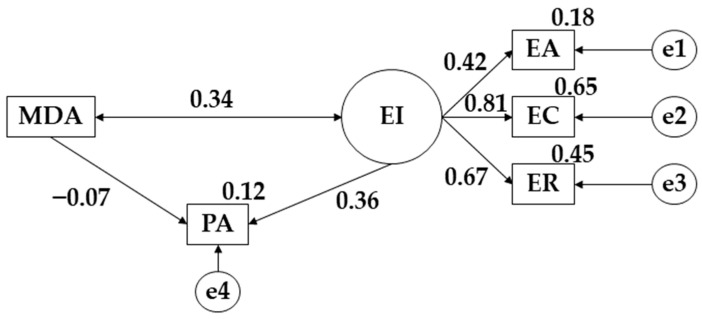
Model proposed for male sample. **Note:** emotional attention (EA); emotional clarity (EC); emotional repair (ER); Mediterranean Diet adherence (MDA); physical activity (PA).

**Figure 4 children-09-01770-f004:**
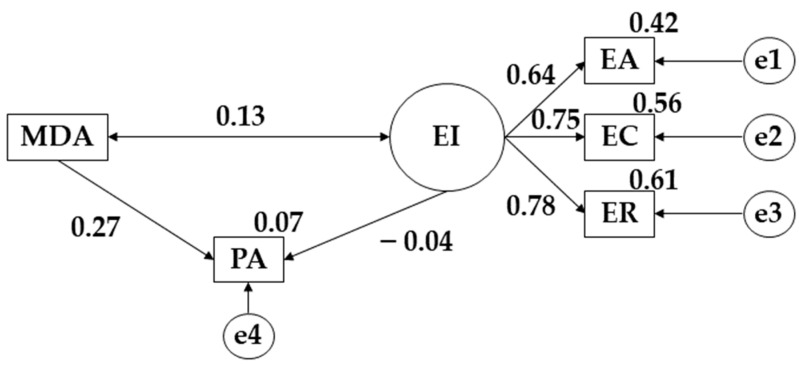
Model proposed for female sample. **Note:** physical activity (PA); Mediterranean Diet adherence (MDA); emotional clarity (EC); emotional attention (EA); emotional repair (ER).

**Table 1 children-09-01770-t001:** Structural Equation Model Developed for the Entire Sample.

Associations between Variables	R.W.				S.R.W.
	Estimates	S.E.	C.R.	*p*	Estimates
EA ș←EI	1.000				0.537
EC←EI	1.151	0.157	7.326	***	0.735
ER←EI	1.181	0.164	7.212	***	0.771
PA←MDA	0.316	0.132	2.392	**	0.143
PA←EI	0.067	0.051	1.322	0.186	0.091
MDA← →EI	0.017	0.005	3.067	**	0.223

**Note 1:** Standardized regression weights (S.R.W); Regression weights (R.W); Estimation error (S.E); Critical ratio (C.R). **Note 2:** Emotional Clarity (EC); Emotional Attention (EA); Emotional Repair (ER); Mediterranean diet adherence (MDA); Physical Activity (PA). **Note 3:** *** *p* ≤ 0.001; ** *p* ≤ 0.05.

**Table 2 children-09-01770-t002:** Structural Equation Model Developed for the Male Population.

Associations between Variables	R.W.				S.R.W.
	Estimates	S.E.	C.R.	*p*	Estimates
EA←EI	1.000				0.421
EC←EI	1.601	0.393	4.074	***	0.807
ER←EI	1.339	0.322	4.161	***	0.668
PA←MDA	−0.129	0.162	−0.796	0.426	−0.069
PA←EI	0.312	0.107	2.923	**	0.363
MDA← →EI	0.020	0.007	2.744	**	0.336

**Note 1:** standardized regression weights (S.R.W); regression weights (R.W); estimation error (S.E); critical ratio (C.R). **Note 2:** emotional Clarity (EC); emotional Attention (EA); emotional repair (ER); Mediterranean Diet adherence (MDA); physical activity (PA). **Note 3:** *** *p* ≤ 0.001; ** *p* ≤ 0.05.

**Table 3 children-09-01770-t003:** Structural Equation Model Developed for female population.

Associations between Variables	R.W.				S.R.W.
	Estimates	S.E.	C.R.	*p*	Estimates
EA←EI	1.000				0.645
EC←EI	0.986	0.154	6.414	***	0.750
ER←EI	0.970	0.152	6.369	***	0.783
PA←MDA	0.704	0.208	3.380	***	0.272
PA←EI	−0.029	0.060	−0.485	0.628	−0.044
MDA← →EI	0.011	0.008	1.332	0.183	0.127

**Note 1:** standardized regression weights (S.R.W); regression weights (R.W); estimation error (S.E); critical ratio (C.R). **Note 2:** emotional attention (EA); emotional clarity (EC); emotional repair (ER); Mediterranean Diet adherence (MDA); physical activity (PA). **Note 3:** *** *p* ≤ 0.001.

## Data Availability

The data used to support the findings of current study are available from the corresponding author upon request.

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
