# Peer review of "Emotional Intelligence, Physical Activity Practice and Mediterranean Diet Adherence-An Explanatory Model in Elementary Education School Students"

_children, 2022, doi:10.3390/children9111770_

Round 1

Reviewer 1 Report

Title: Emotional Intelligence, Physical Activity Practice and Mediterranean Diet Adherence. An Explanatory Model in Elementary Education School Students.

 Article Type: Article

 Summary

 In this study, the authors examined the relationship between adherence to the Mediterranean diet, emotional intelligence and physical activity practice in a sample of 293 elementary school students. Participants completed the sociodemographic questionnaire, the Trait Meta Mood Scale (TMMS-24), the KIDMED questionnaire and the Physical Activity Questionnaire for Older Children. The results indicated that sex plays a fundamental role in adolescence, being a key factor that influences an active and healthy lifestyle.

 Points and suggestions

 Please add more information regarding the results in abstract.

Please add a conclusion to the abstract.

Please add the better keywords (the words which are important but not in the title)

The introduction is very elementary and there is no coherence between the research variables. The research gap and the significance of the study are not well defined.

How the sample size was calculated?

Results section is good but discussion need a more attention for better justification of the results.

Author Response

REVIEWER 1

Comment 1

Summary

In this study, the authors examined the relationship between adherence to the Mediterranean diet, emotional intelligence and physical activity practice in a sample of 293 elementary school students. Participants completed the sociodemographic questionnaire, the Trait Meta Mood Scale (TMMS-24), the KIDMED questionnaire and the Physical Activity Questionnaire for Older Children. The results indicated that sex plays a fundamental role in adolescence, being a key factor that influences an active and healthy lifestyle.

Response 1

Thank you very much for your comment. In this case you have described very well how the study has developed.

Comment 2 

Points and suggestions

Please add more information regarding the results in abstract.

Please add a conclusion to the abstract.

Please add the better keywords (the words which are important but not in the title)

Response 2

Thank you very much for your comments for improvement. In this case, a section on conclusions has been added to the abstract and the results obtained have been reformulated. New keywords have also been proposed.

Comment 3

The introduction is very elementary and there is no coherence between the research variables. The research gap and the significance of the study are not well defined.

How the sample size was calculated?

Response 3

Thank you very much for your various comments. In this case, the introduction has been extended to provide more depth and a better connection between the variables that make up the study. Regarding the sample size, this had not been recorded in the corpus of the article, which is why it has been added in section 2.1 entitled design and participants.

Comment 4

Results section is good but discussion need a more attention for better justification of the results.

Response 4

Thank you very much for your comment. In this case a new paragraph has been added to improve the contextualisation.

Reviewer 2 Report

Thank you for the opportunity to review this study.

I consider the manuscript appropriate to the scope of this Journal, but before acceptance, I made some minor suggestions as follows below:

Please, remove the dot at the end of the title of the manuscript;

Introduction:

I suggest rewriting the first two sentences as follows:

Adolescence is currently a period of human growth and development that occurs between childhood and adulthood [1]. This stage is one of the most crucial times, not only because of the physical changes that occur, but also because of the psychological and social changes that it entails [2].

Adolescence occurs between childhood and adulthood [1], with crucial physical, psychological, and social changes [2].

Despite I like the way how the hypotheses were stated, I think including males compared to females and females compared to males will bring a stronger perspective than the currents. 

Material and Methods:

The term “descriptive” is repeated in the first sentence;

In this phrase “the distribution of the sample is homogeneous, with 50.2% of the sample being male 98 (n=147) and 49.8% female (n=146)” what was the p-value of this comparison??? Please, include it in the text. 

I understand the choice of this statistical analysis is the major strength of the study and I did not see it at the end of the study, but only limitations. Please, prioritize the strengths of the study rather than the limitations.

Author Response

REVIEWER 2

Thank you for the opportunity to review this study.

I consider the manuscript appropriate to the scope of this Journal, but before acceptance, I made some minor suggestions as follows below:

Comment 1

Please, remove the dot at the end of the title of the manuscript;

Response 1

Comment 2

Introduction:

I suggest rewriting the first two sentences as follows:

Adolescence is currently a period of human growth and development that occurs between childhood and adulthood [1]. This stage is one of the most crucial times, not only because of the physical changes that occur, but also because of the psychological and social changes that it entails [2].

Adolescence occurs between childhood and adulthood [1], with crucial physical, psychological, and social changes [2].

Response 2

Thank you very much for your comment. The sentence has been rewritten in response to your comments.

Comment 3

Despite I like the way how the hypotheses were stated, I think including males compared to females and females compared to males will bring a stronger perspective than the currents. 

Response 3

Thank you very much for your suggestions for improvement. Your suggestion for improvement has been added to the research hypotheses.

Comment 4

Material and Methods:

The term “descriptive” is repeated in the first sentence;

In this phrase “the distribution of the sample is homogeneous, with 50.2% of the sample being male 98 (n=147) and 49.8% female (n=146)” what was the p-value of this comparison??? Please, include it in the text. 

Response 4

Thank you very much for your comment. In this case the word "descriptive" has been removed. Regarding the comparison of the sex of the participants, there is no p-value, as this is a descriptive analysis aimed at stating the number of men and women in the research sample.

Comment 5

I understand the choice of this statistical analysis is the major strength of the study and I did not see it at the end of the study, but only limitations. Please, prioritize the strengths of the study rather than the limitations.

Reponse 5

Thank you very much for your comment. In response to your comment, the requested comment has been added to this section.

Reviewer 3 Report

The manuscript is very interesting, with very interesting statistic analysis too but the manuscript  needs minor corrections.

1.    The inclusion of research hypotheses reinforces the statistical analysis.
2.    2. Materials and Methods; 2.1 Design and Participants – the authors should add exclusion criteria. Minors participated in the study did the authors obtain written consent to participate in the study?
3.    All figures - please correct, it is not very readable.
4.    Verse 266, Figure 3 – should be Figure 4.
5.    The authors should use only limitations in manuscript, not ”Limitations and Future Perspectives”.

Author Response

REVIEWER 3

The manuscript is very interesting, with very interesting statistic analysis too but the manuscript  needs minor corrections.

Comment 1

The inclusion of research hypotheses reinforces the statistical analysis.
Response 1

Thank you very much for your comment. In this case the hypotheses have been proposed to give the study more consistency.

Comment 2

Materials and Methods; 2.1 Design and Participants – the authors should add exclusion criteria. Minors participated in the study did the authors obtain written consent to participate in the study?

Response 2

Thank you very much for your comment. In this case the exclusion criteria have been added. In addition, all participants have been authorised by their legal guardians to participate in this study. Regarding the ethical aspects, the research has followed the principles established in the Helsinki declaration and has been approved and supervised by two ethics committees of the University of Granada.

Comment 3

All figures - please correct, it is not very readable.
Response 3

Thank you very much for your suggestion. In this case all the figures have been reworked. In addition, the correct minus symbol has been added instead of the hyphen.

Comment 4

Verse 266, Figure 3 – should be Figure 4.
Response 4

Thank you very much for your comment. The error you have pointed out has been amended.

Comment 5

The authors should use only limitations in manuscript, not ”Limitations and Future Perspectives”.

Response 5

Thank you very much for your suggestion. In this case it has been removed from Future Perspectives.

Round 2

Reviewer 1 Report

Thank you to the efforts of the authors for the revising made, however, I still believe this article has a lot of basic problems and so cant be consider for publication in the Journal.

Author Response

Thank you very much for your comments. 
